# Deciphering the Disaggregation Mechanism of Amyloid Beta Aggregate by 4-(2-Hydroxyethyl)-1-Piperazinepropanesulfonic Acid Using Electrochemical Impedance Spectroscopy

**DOI:** 10.3390/s21030788

**Published:** 2021-01-25

**Authors:** Hien T. Ngoc Le, Sungbo Cho

**Affiliations:** 1Department of Electronic Engineering, Gachon University, Seongnam-si, Gyeonggi-do 13120, Korea; ltnh1809@gachon.ac.kr; 2Department of Health Sciences and Technology, GAIHST, Gachon University, Incheon 21999, Korea

**Keywords:** Alzheimer’s disease, amyloid beta aggregate, 4-(2-hydroxyethyl)-1-piperazinepropanesulfonic acid, impedimetric immunosensor, electrochemical impedance spectroscopy, kinetic disaggregation of protein

## Abstract

Aggregation of amyloid-β (aβ) peptides into toxic oligomers, fibrils, and plaques is central in the molecular pathogenesis of Alzheimer’s disease (AD) and is the primary focus of AD diagnostics. Disaggregation or elimination of toxic aβ aggregates in patients is important for delaying the progression of neurodegenerative disorders in AD. Recently, 4-(2-hydroxyethyl)-1-piperazinepropanesulfonic acid (EPPS) was introduced as a chemical agent that binds with toxic aβ aggregates and transforms them into monomers to reduce the negative effects of aβ aggregates in the brain. However, the mechanism of aβ disaggregation by EPPS has not yet been completely clarified. In this study, an electrochemical impedimetric immunosensor for aβ diagnostics was developed by immobilizing a specific anti-amyloid-β (aβ) antibody onto a self-assembled monolayer functionalized with a new interdigitated chain-shaped electrode (anti-aβ/SAM/ICE). To investigate the ability of EPPS in recognizing AD by extricating aβ aggregation, commercially available aβ aggregates (aβ_agg_) were used. Electrochemical impedance spectroscopy was used to probe the changes in charge transfer resistance (R_ct_) of the immunosensor after the specific binding of biosensor with aβ_agg_. The subsequent incubation of the aβ_agg_ complex with a specific concentration of EPPS at different time intervals divulged AD progression. The decline in the R_ct_ of the immunosensor started at 10 min of EPPS incubation and continued to decrease gradually from 20 min, indicating that the accumulation of aβ_agg_ on the surface of the anti-aβ/SAM/ICE sensor has been extricated. Here, the kinetic disaggregation rate *k* value of aβ_agg_ was found to be 0.038. This innovative study using electrochemical measurement to investigate the mechanism of aβ_agg_ disaggregation by EPPS could provide a new perspective in monitoring the disaggregation periods of aβ_agg_ from oligomeric to monomeric form, and then support for the prediction and handling AD symptoms at different stages after treatment by a drug, EPPS.

## 1. Introduction

The deposition of the amyloid-β (aβ) protein in the cortex of the brain is reported as a leading cause of Alzheimer’s disease (AD), the most common form of dementia [1]. According to the amyloid cascade hypothesis, aβ peptides tend to aggregate and coalesce into toxic insoluble oligomers and fibrils in plaques, which may cause decline in synaptic function and nerve cell damage [2,3].

Aggregation of aβ gradually increases throughout the brain, which leads to neurodegeneration, starting from mild to moderate symptoms in earlier Alzheimer’s stages, and finally dementia [3,4]. Many efforts have been made to control and prevent Alzheimer’s progression by inhibiting and reducing the aggregation of aβ. For instance, laser-induced destruction [5], ultrasonic treatment [6], temperature-induced dissociation [7], pulsed radio-frequency cold atmospheric plasma jet [8], and drug treatments [9,10] have all been used to eliminate the aggregation of aβ. Among these approaches, drug treatment is the most-studied means of reducing or eliminating aβ aggregation, as these agents provide the most practical strategy to target the aβ aggregates within the body. Some natural chemical compounds such as theaflavins from black tea, epigallocatechin 3-gallate from green tea, and components from coffee showed an inhibitory effect on the aggregation of aβ [11,12,13]. However, the precise mechanism of interaction between these agents and aβ aggregates remains uncertain, and none of these have been used to treat AD clinically.

Recently, 4-(2-hydroxyethyl)-1-piperazinepropanesulfonic acid (EPPS) has been acknowledged as a promising drug candidate for dissociating and converting aβ aggregates into monomers. A study on a mouse model of Alzheimer’s aβ plaques found that EPPS can contribute to the breakdown of the plaques, which can alleviate some of the AD symptoms in mice [14]. Despite its significance, the detailed mechanism of EPPS on the disaggregation of aβ aggregates has not been thoroughly elucidated.

Electrochemical immunosensors offer rapid response, high sensitivity, and selective detection of aβ at very low concentrations [15,16,17,18]. In this report, we created an impedimetric immunosensor for the diagnosis of aβ aggregates (aβ_agg_) and aβ peptide (aβ_pep_). It was produce by the immobilization of a specific anti-aβ antibody on a self-assembled monolayer functionalized with a new interdigitated chain-shaped electrode (anti-aβ/SAM/ICE). This was then used to monitor the effect of EPPS on the disaggregation aβ_agg_. After the anti-aβ/SAM/ICE biosensor bound with aβ_agg_, forming the aβ_agg_ complex, the aβ_agg_ complex was then incubated with a specific concentration of EPPS at different time intervals to divulge the disaggregation progress of aβ_agg_. Electrochemical impedance spectroscopy (EIS) was used as a tool to confirm the changes in charge transfer resistance (R_ct_) of the immunosensor after forming the complex and EPPS treatment. We established a new method for studying the mechanism of EPPS-induced kinetic disaggregation of aβ_agg_ by analyzing the EIS results from this study, leading to new insights in the diagnostics and treatment of AD symptoms at various levels.

## 2. Materials and Methods

### 2.1. Materials

AggreSureᵀᴹ Amyloid-β 1–42 aggregate (aβ_agg_) is bought from AnaSpec (Fremont, CA 94555, USA). Anti-aβ antibody ab126649 (anti-aβ), Amyloid beta peptide 1–42, human, ab120301 (aβ_pep_) are bought from Abcam (Seoul, Korea).

De-ionized (DI) water is gained from Milli-Q system. Phosphate buffer saline (PBS, pH 7.4) is received from Tech-Innovation.

4-(2-hydroxyethyl)-1-piperazinepropanesulfonic acid (EPPS, BioXtra, ≥99.5% [titration]), EDC hydrochloride (crystalline, ≥98.0%), 6-mercaptohexanoic acid (MHA, 90%), 1-hydroxy-2,5-pyrrolidinedione (NHS, 98.0%), potassium ferrocyanide/ferricyanide ([Fe(CN)_6_]^3−/4−^), bovine serum albumin (BSA, 0.5% in 1 × PBS, pH 7.4) are obtained from Sigma Aldrich (Seoul, Korea).

### 2.2. Development of the Immunosensor for Monitoring the Impact of EPPS on the Disaggregation of aβ_agg_

On a glass transparency substrate (13.5 × 16.0 × 0.5 mm), an interdigitated chain-shaped electrode (ICE) is made. Ti and Au have been deposited in an electron beam-evaporator with electrodes 25 nm and 50 nm of thickness. The lift-off procedure then formed the coupled electrode finger consisting of 5 μm distance and width for working and reference electrodes. The ICE was then cleaned with 100% ethanol, DI water, and dried in nitrogen gas flow to prepare for the development of the immunosensor. Schematic of the fabrication and microscope image of ICE are displayed in Figure 1.

The development of the immunosensor for diagnosis of aβ_agg_ is displayed in Figure 2. A self-assembled monolayer (SAM) has been mounted on the purified ICE gold surface in 100 mM MHA for 24 hours at room temperature. A solution of 75 mM EDC and 5 mM NHS was then incubated with the SAM-modified electrode to activate the carboxyl group for antibody binding. Then, ten anti-aβ antibody microliters (100 μg·mL^−1^) in PBS (pH ~ 7.4) are dripped off into a modified electrode and incubated for 1 hour in a moist compartment at 4 °C. Therefore, the EDC/NHS-activated SAM molecules on the modified electrodes were attached to the anti-aβ antibody through the amino group. After that, nonspecific adsorption was blocked by incubating with BSA (0.5% in 1 × PBS, pH 7.4) for 20 min at 4 °C to develop the anti-aβ/SAM/ICE biosensor. The developed anti-aβ/SAM/ICE biosensor was then incubated with aβ_agg_ (in PBS (pH ~ 7.4)) for 20 min at room temperature for the recognition of aβ_agg_.

To monitor the effect of EPPS on the kinetics disaggregation of aβ_agg_, the anti-aβ/SAM/ICE biosensor was incubated with aβ_agg_ (2.5 × 10^−3^ mg·mL^−1^) for 20 min to enable a specific binding between the antibody and antigen, called the aβ_agg_ complex. Subsequently, the aβ_agg_ complex was incubated with 20 mM·L^−1^ of EPPS for various time periods to study the disaggregation process of aβ_agg_ by electrochemical impedance measurement.

### 2.3. EIS Measurements and Functional Group Characterization

EIS was performed using EC-Lab (SP-200, Bio-Logic, France) in 5 mM [Fe(CN)_6_]^3−/4−^ in PBS (pH 7.4) at each step of the anti-aβ/SAM/ICE biosensor development and various time intervals of the disaggregation process of aβ_agg_. The EIS data was obtained by putting on a 10 mV of sinus amplitude which satisfy the linear properties of electrical response of the object between the working and reference electrode from 100 mHz to 1 MHz of range frequency. Z-Fit software (EC-Lab, Bio-Logic, France) was used to fit the obtained EIS data through by the Randle’s equivalent circuit.

Fourier-transform infrared (FT-IR) spectra was characterized by using Jasco-4600 FTIR (USA) spectrometer to confirm the functional groups for the fabrication steps of the anti-aβ/SAM/ICE biosensor.

## 3. Results and Discussion

### 3.1. Investigating the Development of Impedimetric Immunosensor

As shown in Figure 3a, EIS measurements were carried out to determine the Nyquist-plots with the imaginary impedance component (-Im(Z)) plotted with the actual impedance component (Re(Z)) to confirm the successful modification of substantial ingredients to the electrode surface at different stages of the development the anti-aβ/SAM/ICE immunosensor.

The equivalent circuit model of Randle insert into Figure 3a was used to fit the Nyquist plot data, showing three EIS parameters, including the C (capacitance for the electrode-solution interface element), R_ct_ (interfacial charge-transfer resistance that corresponds to the semicircular diameter in the Nyquist plot), and R_s_ (solution resistance).

The obtained fitting value of R_ct_ from Figure 3a was displayed in Figure 3b and Table 1. After SAM deposition that can be attributed to the −COO‾ group and the carbon chain SAM which prevents the transfer of the negative charged [Fe(CN)_6_]^3−/4−^ redox couple to the electrode surface, the value of R_ct_ has increased dramatically. Then, R_ct_ decreased after the EDC–NHS layer was deposited on the SAM-modified electrode, suggesting that the −COO‾ groups of SAM were activated with the EDC–NHS carbodiimide coupler, the formation of succinimide ester on the modified electrode surface of SAM allows fast electron transfer between the electrode and the interface [19]. The two continuous immobilization steps of anti-aβ and BSA showed an increase in R_ct_ due to the binding of a specific material on the sensor surface which enhanced its surface resistance [20], demonstrating that the surface of the biosensor was covered by specific materials.

Fourier-transform infrared (FT-IR) spectra was also used to confirm the fabrication of the anti-aβ/SAM/ICE biosensor, and the results were displayed in Figure 4. The FT-IR result of the SAM-modified ICE showed the typical peaks near 1601 cm^−1^ of carbonyl from the carboxylic acid moiety on the alkanethiols of SAM [21], indicating the immobilization of SAM on the ICE. FT-IR result of the immobilized anti-aβ on the EDC-NHS coupler activated SAM-modified ICE (blue curve in Figure 4) showed the typical peaks of protein at near 3374 cm^−1^ of N-H stretching of amide II [22], 2925 and 2849 cm^−1^ of C-H stretching of -CH_2_ of protein [23,24], 920 cm^−1^ of C-O stretching [25], and 602 cm^−1^ of N-H deformation [22], which demonstrated the successful development of the anti-aβ/SAM/ICE biosensor.

### 3.2. Monitoring the Effect of EPPS on aβ Disaggregation

EPPS was used to target the disaggregation of aβ_agg_ into its monomeric form. To confirm whether EPPS had an effect on the disaggregation of aβ_agg_, EIS was used to explore the change in R_ct_ of the specific anti-aβ/SAM/ICE biosensor for recognizing aβ during incubation with EPPS at different time intervals. 

Firstly, the impact of EPPS on the anti-aβ antibody was studied. EIS measurements of the anti-aβ/SAM/ICE biosensor after incubation with 20 mM·L^−1^ EPPS at different time intervals are displayed in Figure 5. The Nyquist plots in Figure 5a did not change significantly after treatment with EPPS. Using Randle’s equivalent circuit to fit the EIS data of Nyquist plots in Figure 5a, the R_ct_ value was generated that serve to normalize the response of the anti-aβ/SAM/ICE biosensor to the EPPS treatment at 10, 20, 30, 40, 50 min in Figure 5b, where R_0_ represents the R_ct_ value of the biosensor with the immobilization of anti-aβ antibody, and R_a_ represents the R_ct_ value of the anti-aβ/SAM/ICE biosensor within the treatment of EPPS. The normalized value (R_a_ − R_0_)/R_0_ of the anti-aβ/SAM/ICE biosensor in Figure 5b did not change significantly after treatment with EPPS at various time periods, indicating that EPPS did not affect the anti-aβ antibody. The fitting EIS data were shown in Table 2.

Gel electrophoresis method [26] was used to confirm the aggregation state of aβ in this manuscript. As seen in Figure 6, the gel electrophoresis result of aβ_agg_ showed the molecular-weight ranging of 11 kDa for oligomer form, which corresponds to a smear band containing monomers and low-molecular-weight oligomers ranging between 4 and 18 kDa in size of aβ oligomers [27], demonstrating the oligomeric form of aβ_agg_ in this research.

The process of disaggregation of aβ_agg_ by EPPS at different time intervals is described in Section 2.2. Initially, the anti-aβ/SAM/ICE biosensor was incubated with aβ_agg_ (2.5 × 10^−3^ mg·mL^−1^) for 20 min in the absence of EPPS and measured via EIS to examine the aβ recognition of the biosensor. EIS measurements showed an increase in the diameter and height of the semicircle in the Nyquist plot, corresponding to the enhanced R_ct_ after the anti-aβ/SAM/ICE biosensor was incubated with aβ_agg_, which is marked by 0 min in Figure 7a. This enhancement of R_ct_ is ascribed to the hindrance of electron transfer between the [Fe(CN)_6_]^3−/4−^ redox couple and the electrode surface due to successful specific binding between anti-aβ antibody and aβ_agg_, called aβ_agg_ complex on the surface of the sensor. The biosensor with the aβ_agg_ complex was subsequently incubated for different time periods (10, 20, 30, 40, and 50 min) with 20 mM·L^−1^ of EPPS and was used for EIS measurements. As shown in Figure 7a, after 10 min of incubation in EPPS, the R_ct_ of the biosensor with the aβ_agg_ complex in the Nyquist plot showed an increase in the height and diameter of the semicircle, and the Nyquist plot remained the same for the next 20 min of incubation with EPPS, which indicated that the [Fe(CN)_6_]^3−/4−^ ion transport towards the electrode surface was blocked due to excessive aβ on the electrode surface. These results implied that the disaggregation of aβ_agg_ started after 10 min of incubation with EPPS and was continuously maintained after 20 min of incubation. This disaggregation process converted the aβ_agg_ into small aβ monomers, and still retained a certain amount of aβ_agg_, creating an abundant compound of aβ monomers and aβ_agg_ on the biosensor surface, as delineated in Figure 8a, which hindered the transportation of [Fe(CN)_6_]^3−/4−^ redox couple to the electrode surface, leading to an increase in R_ct_. Figure 7a showed the decline in R_ct_ after treatment with EPPS at 30 min, which manifested the disaggregation of the remaining aβ_agg_ into several monomeric aβ molecules on the sensor surface, suggesting that the movement of ion [Fe(CN)_6_]^3−/4−^ toward the electrode surface was facilitated; the impeded ion [Fe(CN)_6_]^3−/4−^ movement was caused by the presence of a dense aβ_agg_ that was reduced by converting the aβ_agg_ molecules into aβ monomers, leading to the expansion of spacing between the specific aβ molecules on the electrode surface, resulting in facilitating transport of redox couple [Fe(CN)_6_]^3−/4−^. This decline of R_ct_ continued to occur at 40 and 50 min during EPPS treatment, which indicated the continuous disaggregation of aβ_agg_. A graphical description of the disaggregation process of aβ_agg_ by EPPS at various time intervals is shown in Figure 8a. The response of aβ_agg_ to EPPS treatment at time periods was evaluated by normalizing the R_ct_ value of Figure 7a and plotted against time, as (R_b_ − R_i_)/R_i_ in Figure 7b, where R_i_ represents the R_ct_ of the biosensor with the immobilization of aβ_agg_, and R_b_ represents the R_ct_ of the biosensor with the immobilization of aβ_agg_ within the EPPS treatment. The normalized value (R_b_ − R_i_)/R_i_ in Figure 7b increased after 10 min of EPPS treatment, remained stable after 20 min, and gradually decreased over the 30, 40, and 50 min points, which indicated the disaggregation process of aβ_agg_ at different time intervals starting at 10 min and confirmed the effect of EPPS on aβ_agg_ disaggregation, corresponding to the results and explanation in Figure 7a. A detailed graphical description of the disaggregation process of aβ_agg_ by EPPS at various time intervals is provided in Figure 8a. The fitting EIS data were shown in Table 3.

The effect of EPPS on the aβ 1–42 peptide (aβ_pep_), a long peptide of 1–42 amino acids, was also studied for comparison with the obtained results of aβ_agg_, which are mentioned above. The experiment of aβ_pep_ treatment in EPPS was carried out through similar processes as described in Section 2 with aβ_agg_ replaced by aβ_pep_. Nyquist plot of EIS result in Figure 9a show an increase in R_ct_ of the biosensor anti-aβ/SAM/ICE after 20 min of incubation with aβ_pep_. This increment of R_ct_ is expected due to the repulsion between the negative charge of immobilized aβ_pep_ antigen on the anti-aβ/SAM/ICE biosensor with redox couple, that could prevent the redox couple from reaching the SAM-modified electrode, confirming the successful binding between anti-aβ and aβ_pep_ [28]. The R_ct_ of the biosensor with the immobilized aβ_pep_ remained the same after 10 min of treatment in EPPS, indicating that aβ_pep_ was not affected by EPPS for the initial 10 min treatment. R_ct_ significantly decreased after 20 and 30 min of subsequent EPPS treatment, indicating the facilitating electron transfer at the electrode interface. This decrease of R_ct_ may be attributed to the fragmentation of aβ_pep_ by EPPS, leading to the formation of ion-complexed amyloid beta with [Fe(CN)_6_]^3−/4−^ [29,30], which results in the decreasing of R_ct_ and enhancing of electrochemical property of the sensor at 20 min treatment, and at more treatment time of 30 min [16]. The normalized result (R_b_ − R_i_)/R_i_ is displayed in Figure 9b, where R_i_ represents the R_ct_ of the biosensor with the immobilization of aβ_pep_, and R_b_ represents the R_ct_ of the biosensor with the immobilization of aβ_pep_ within the EPPS treatment, which corresponds with the data of Nyquist plots in Figure 9a. The (R_b_ − R_i_)/R_i_ was stable after 10 min, and rapidly decreased over the 20 and 30 min points following EPPS treatment. Figure 8b shows the effect of EPPS treatment on aβ_pep_, and EIS results are shown in Figure 9. The fitting EIS data were shown in Table 4.

By examining the EIS results of aβ_agg_ and aβ_pep_ in the treatment with EPPS in this study, it is observed that EPPS affects the disaggregation of aβ_agg_ gradually and leads to the establishment of a disaggregation mechanism, as shown in Figure 8, while EPPS causes rapid disruption of aβ_pep_.

### 3.3. Kinetic Disaggregation Rate k

The kinetic disaggregation rate *k* of protein is an important parameter in evaluating the functioning of neurodegenerative diseases [31]. However, there have not been report on the *k* value for aβ_agg_ at present. To estimate the disaggregation rate *k* value for aβ_agg_, the normalized data for the treatments of aβ_agg_ and aβ_pep_ in EPPS were plotted and fitted versus time, as shown in Figure 10. Data for the disaggregation experiment aβ_agg_ represent the best fit of the data to a single-exponential function y = *0.149* − *0.048*exp(*0.038*x), whereas data for the treatment aβ_pep_ in EPPS express a linear fit function y = *0.271* − *0.027*x. According to the studies of kinetic disaggregation of proteins, the fitting data for protein disaggregation has been expressed by an exponential function F = *y*_0_ + *A*exp(−*k*t) [31,32], where *k* represents the disaggregation rate. Consequently, the disaggregation rate *k* of aβ_agg_ was determined to be 0.038 by our method, providing a reference parameter for the disaggregation of aβ_agg_ for future studies.

## 4. Conclusions

A new concept of measuring aβ_agg_ disaggregation following treatment with EPPS via EIS signaling was introduced in this report. An electrochemical impedimetric immunosensor was developed to monitor EIS signals at the disaggregation stages of aβ_agg_ during treatment with EPPS at different time intervals. The obtained EIS results confirmed that the disaggregation of aβ_agg_ by treatment in EPPS occurred after 10 min, and this disaggregation process continued gradually over the next 50 min. A single exponential function for aβ_agg_ disaggregation was also found in this report, with a kinetic disaggregation rate (*k*) value of 0.038. The data indicated the successful application of EIS measurement in monitoring and establishing the mechanism for the impact of EPPS on the disaggregation of aβ_agg_, which may lead to new insights in the diagnosis and treatment of signs of AD at various phases.

## Figures and Tables

**Figure 1 sensors-21-00788-f001:**
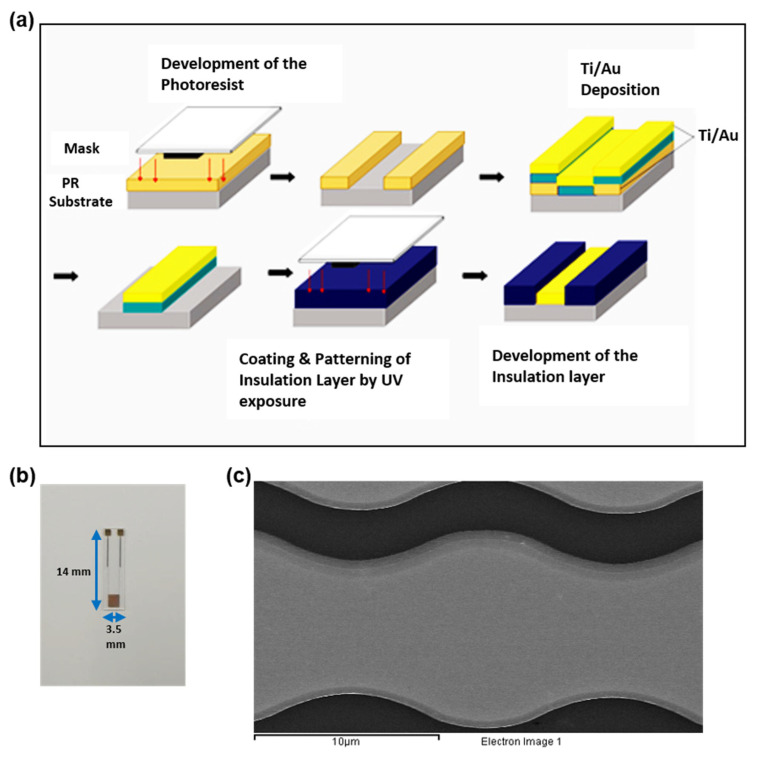
(**a**) Fabrication of ICE, (**b**) Photograph of ICE, and (**c**) Scanning electron microscope image of chain-shaped gold finger of ICE.

**Figure 2 sensors-21-00788-f002:**
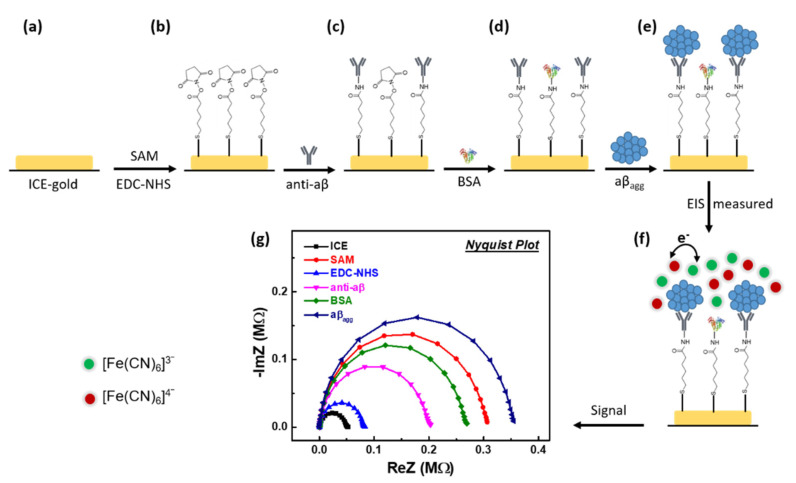
(**a**–**e**) Development procedure the immunosensor anti-aβ/SAM/ICE for the recognition aβ_agg_, (**f**) impedimetric measurement mechanism in the recognition aβ_agg_ by the developed biosensor anti-aβ/SAM/ICE, and (**g**) impedimetric results represent by Nyquist plot at each step of the development anti-aβ/SAM/ICE biosensor in the recognition aβ_agg_.

**Figure 3 sensors-21-00788-f003:**
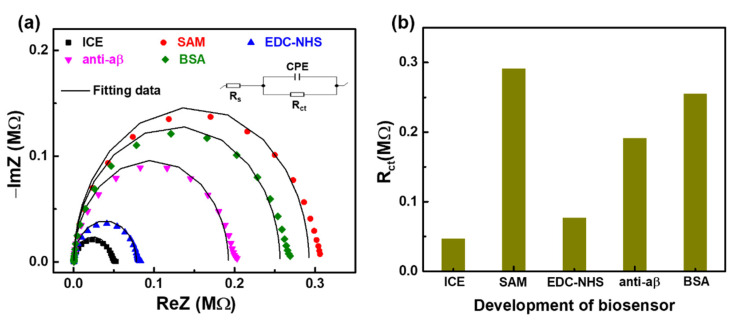
EIS results at each step of the development of biosensor anti-aβ/SAM/ICE display in (**a**) Nyquist plots and (**b**) charge-transfer resistance (R_ct_).

**Figure 4 sensors-21-00788-f004:**
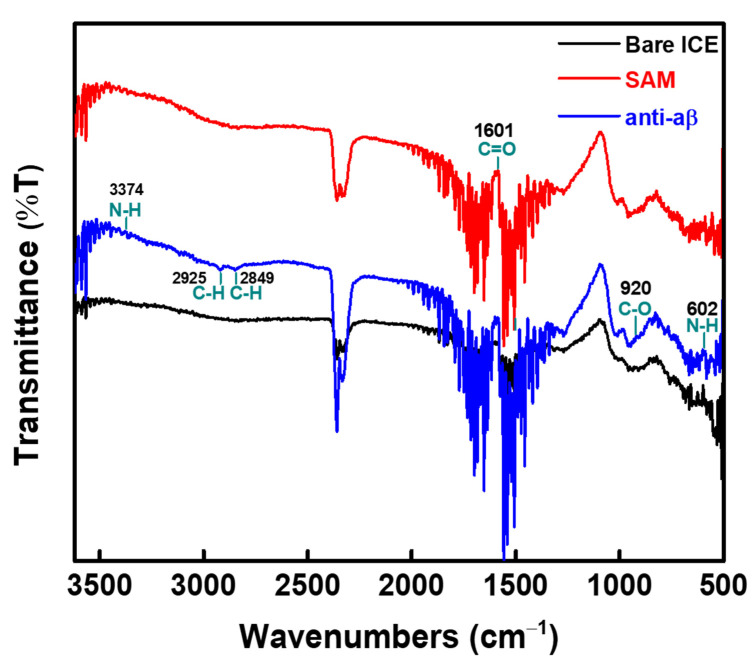
FT-IR results for the fabrication of the anti-aβ/SAM/ICE biosensor.

**Figure 5 sensors-21-00788-f005:**
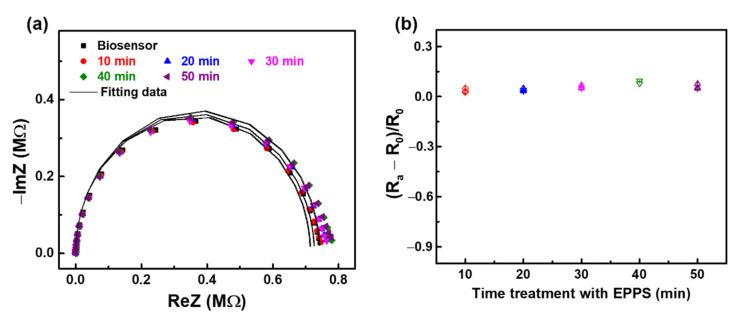
EIS results of the biosensor anti-aβ/SAM/ICE with the treatment in EPPS at different time intervals display in (**a**) Nyquist plots and (**b**) Normalization of R_ct_. (*n* = 3; three data points shown).

**Figure 6 sensors-21-00788-f006:**
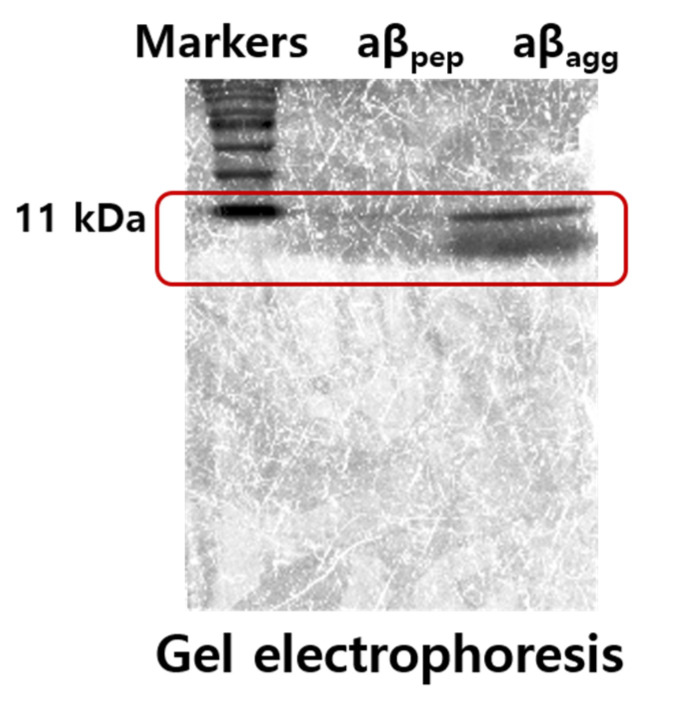
Gel electrophoresis results of aβ_pep_ and aβ_agg_.

**Figure 7 sensors-21-00788-f007:**
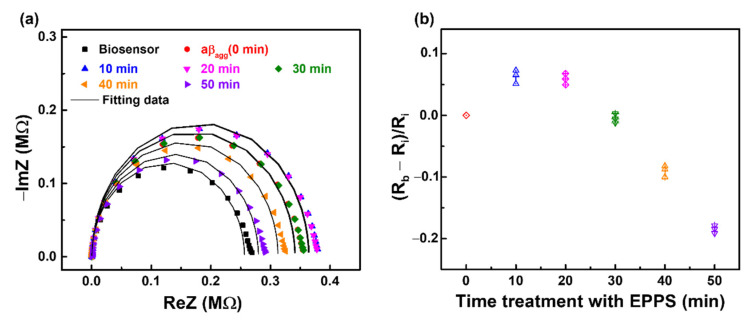
EIS results monitoring the disaggregation aβ_agg_ with the treatment in EPPS at different time intervals display in (**a**) Nyquist plots and (**b**) Normalization of R_ct_. (*n* = 3; three data points shown).

**Figure 8 sensors-21-00788-f008:**
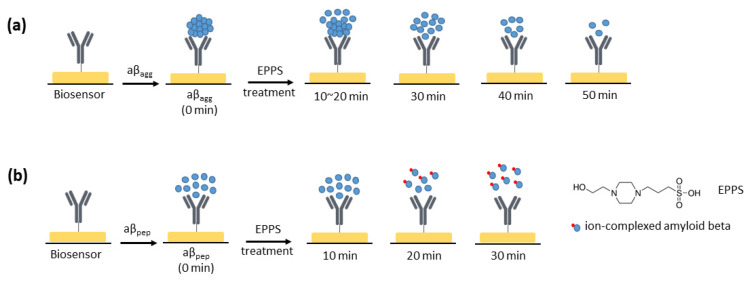
(**a**) Disaggregation mechanism of aβ_agg_ by EPPS, and (**b**) the impact of EPPS on aβ_pep_.

**Figure 9 sensors-21-00788-f009:**
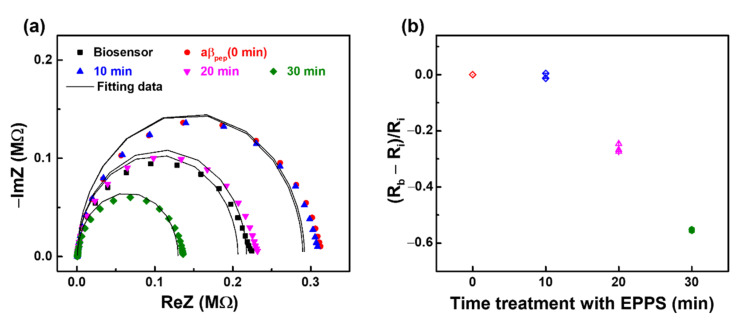
EIS results monitoring the impact of EPPS on the aβ_pep_ at different time intervals display in (**a**) Nyquist plots and (**b**) Normalization of R_ct_. (*n* = 3; three data points shown).

**Figure 10 sensors-21-00788-f010:**
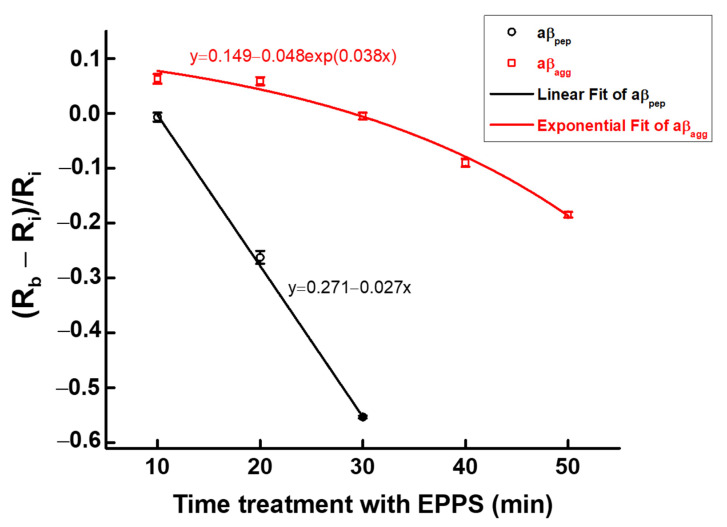
Fitting of two sets of disaggregation data for aβ_agg_ and aβ_pep_, respectively; symbols and bars represent the average and standard deviation of the data (*n* = 3).

**Table 1 sensors-21-00788-t001:** EIS parameters of the bare and modified ICE extrapolated by fitting the measured spectra to the equivalent circuit model shown in Figure 3a.

Electrode	R_s_ (Ω)	C (F)	R_ct_ (Ω)
ICE	529.3	55.37 × 10^−9^	47,294
SAM	522.8	77.78 × 10^−9^	291,956
EDC-NHS	524.9	69.69 × 10^−9^	77,691
anti-aβ	524	75.88 × 10^−9^	191,820
BSA	504.5	76.49 × 10^−9^	255,926

**Table 2 sensors-21-00788-t002:** EIS parameters of the anti-aβ/SAM/ICE biosensor after treatment with EPPS at different time intervals, extrapolated by fitting the measured spectra in Figure 5a to the equivalent circuit model of Randle.

Electrode	R_s_ (Ω)	C (F)	R_ct_ (Ω)
Biosensor	490.9	55.55 × 10^−9^	715,084
10 min	491.9	56.72 × 10^−9^	724,551
20 min	487.5	57.66 × 10^−9^	726,538
30 min	488.7	57.93 × 10^−9^	724,904
40 min	488.5	58.36 × 10^−9^	743,262
50 min	486.3	58.83 × 10^−9^	730,896

**Table 3 sensors-21-00788-t003:** EIS parameters of the anti-aβ/SAM/ICE biosensor after incubating with aβ_agg_, and then conduct to treatment with EPPS at different time intervals, extrapolated by fitting the measured spectra in Figure 7a to the equivalent circuit model of Randle.

Electrode	R_s_ (Ω)	C (F)	R_ct_ (Ω)
Biosensor	504.8	76.10 × 10^−9^	228,331
aβ_agg_ (0 min)	519.2	76.29 × 10^−9^	331,631
10 min	517.4	75.98 × 10^−9^	366,841
20 min	517.1	75.89 × 10^−9^	363,145
30 min	519.1	75.08 × 10^−9^	334,918
40 min	519.5	74.81 × 10^−9^	307,364
50 min	517.6	74.71 × 10^−9^	274,678

**Table 4 sensors-21-00788-t004:** EIS parameters of the anti-aβ/SAM/ICE biosensor after incubating with aβ_pep_, and then conduct to treatment with EPPS at different time intervals, extrapolated by fitting the measured spectra in Figure 9a to the equivalent circuit model of Randle.

Electrode	R_s_ (Ω)	C (F)	R_ct_ (Ω)
Biosensor	633.5	91.3 × 10^−9^	205,978
aβ_pep_ (0 min)	635	92.96 × 10^−9^	289,018
10 min	632.6	92.17 × 10^−9^	289,110
20 min	629.8	90.69 × 10^−9^	209,893
30 min	639.9	88.06 × 10^−9^	129,281

## Data Availability

Not applicable.

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
