# Peer review of "Deciphering the Disaggregation Mechanism of Amyloid Beta Aggregate by 4-(2-Hydroxyethyl)-1-Piperazinepropanesulfonic Acid Using Electrochemical Impedance Spectroscopy"

_sensors, 2021, doi:10.3390/s21030788_

Round 1

Reviewer 1 Report

The paper describes an anti-body detection method for Alzheimer’s disease diagnostics. The method detects the level of ab protein using EIS to detect changes in charge transfer resistance of the immunosensor after forming the complex and EPPS treatment

The background is professionally written, and the experiment setup and fabrication, as well as the working mechanism of the biosensor, is well presented.

The paper also investigates the effect of EPPS treatment and discuss its impact on ab aggregation over time.

The reviewer has the following minor suggestions:

There are many short terms, it is suggested to explain them as clear as possible.

ICE is not expressed the 1st time used; it is in Fig.1 instead.

Fig.2 seems to suggest EIS measurement is taken only after aB_agg added, but EIS measurements were in fact taken every step?

Reviewer 2 Report

In this work, Le and Cho report the electrochemical impedimetric immunosensor for amyloid β (aβ) diagnostics was developed, which can investigate the mechanism of aβ aggregates disaggregation by EPPS. It is an interesting work where authors provide a new strategy for designing electrochemical impedimetric immunosensor by immobilizing a specific anti-amyloid-β (aβ) antibody onto a self-assembled monolayer functionalized with a new interdigitated chain-shaped electrode (anti-aβ/SAM/ICE). In my opinion, the manuscript can be published in this journal, after some points should be clarified. The following issues should be addressed.

Major issues

- In the last sentence of the abstract, the authors said that the electrochemical sensor developed by the authors will be a new strategy in diagnosis and dealing with symptoms of AD at different stages. However, this work is not an experimental result related to diagnosis, and it is expected to be used to monitor the aβ elimination effect of drugs such as EPPS. Therefore, justify that the results of this experiment will be helpful in diagnosis.

- In Figure 2 results and corresponding discussion, the process of manufacturing the anti-aβ/SAM/ICE bio-sensor should be cross-checked by a method other than an electrochemical method.

- Were the aβ aggregates used in the experiment in the form of oligomers or fibrils? Authors must specify the type of aggregates. It seems to be an oligomer from the schematic illustrations in Figures, but it must be proved by presenting visual data.

Reviewer 3 Report

Herein, author examined the Amyloid Beta Aggregate by 4-(2-Hydroxyethyl)-1-Piperazinepropanesulfonic Acid Using Electrochemical Impedance Spectroscopy. The concept is good. The manuscript is not well written. Hence, I don’t recommend the acceptance of this manuscript. Author should revise the manuscript by considering the following comments.

  1. The English grammar, typos, uses of prepositions, etc. should be checked carefully. I strongly recommend to revise the manuscript by English editing service.
  2. It seems Figure 1c is obtained for the EDX measurements. Please add a clear image without showing the spectrum 2 inside the image. Also, I recommend to increase the font size or make a clearer image of Fig. 1a, currently, the image resolution is poor.
  3. In the section 2.2, author should add the solution pH, dipping time, temperature for antibody and BSA immobilization and aβagg attachment.
  4. Section 2.3, why author choose 10 mV as input voltage (I think 10 mV is sinusoidal wave amplitude)? What is applied potential for the EIS measurements?
  5. The representation of ideal Nyquist plot should be drawn using same resistance range values in both X- and Y-axes. I wonder why there is no Warburg diffusion resistance observed in EIS spectra for such a standard redox probe. Author should provide both experimental and fitting EIS data in all the EIS plots.
  6. Author should explain the reason the for the dramatic decrease of Rct after activation, and increases of Rct after antibody and BSA immobilization with relevant references.
  7. I page # 6, author explained ‘’Nyquist plots of EIS results in Figure 7a show an increase in Rct of the anti-aβ/SAM/ICE biosensor after 20 min of incubation with aβpep, confirming the successful formation of the binding anti-aβ and aβpep, blocking the transfer of the redox couple [Fe(CN)6]3‾/4‾ toward the electrode surface’’. Please explain the reason of blocking the electron transfer.
  8. I page # 7, Rct significantly decreased after 20 and 30 min of subsequent EPPS treatment, which was related to the enhancement of the negative charge at the electrode interface providing higher electron transfer pathways’’. In my opinion the increase of the negative charge at the electrode interface should provide lower electron transfer pathways due to the electrostatic repulsion with the negatively charge redox probe. Please clarify this.
  9. Why the Rct is lower than Rct of the biosensor after 30 min in Fig. 7a.
  10. The characterization of the sensors in each fabrication steps was not characterized well. It is anticipated to see some analyses to confirm the successful fabrication of electrode such as CV, FTIR, contact angle, AFM etc.

Round 2

Reviewer 2 Report

The authors sincerely answered the questions.

Author Response

Thanks for reviewer's acceptance.

Reviewer 3 Report

 I am still critical about the reply of some comments which author did not addressed correctly.

  1. For specific comments (Section 2.3, why author choose 10 mV as input voltage (I think 10 mV is sinusoidal wave amplitude)? What is applied potential for the EIS measurements?), generally 0.3V bias potential is used for the ferricyanide redox couple for  EIS measurement. Does author used this potential or open circuit condition?
  2. The representation of ideal Nyquist plot should be drawn using same resistance range values in both X- and Y-axes. Author must provide both experimental and fitting EIS data in all the EIS plots (providing a table is not enough).
  3. In response of the comments (Author should explain the reason the for the dramatic decrease of Rct after activation, and increases of Rct after antibody and BSA immobilization with relevant references  and In page # 6, author explained ‘’Nyquist plots of EIS results in Figure 7a show an increase in Rct of the anti-aβ/SAM/ICE biosensor after 20 min of incubation with aβpep, confirming the successful formation of the binding anti-aβ and aβpep, blocking the transfer of the redox couple [Fe(CN)6]3‾/4‾ toward the electrode surface’’. Please explain the reason of blocking the electron transfer) author replied with some wordy sentence such as ''increase or decrease… is due to the formation of an impediment layer between anti-aβ and aβpep through by the specific binding between antibody (anti-aβ) and antigen (aβpep)''. This is not a scientific explanation. Obviously, the explained conclusions can be concluded after explained the proper scientific reason in view of electrochemistry. Please consider the variation of charge issue of electrode (SAM layer) and the charge perturbation issue induced by the protein. This can hinder or varied the diffusion of negatively charged redox couple, concurrently the Rct increase or decrease (specific case).
